# Effect of V on the Precipitation Behavior of Ti−Mo Microalloyed High-Strength Steel

**DOI:** 10.3390/ma15175965

**Published:** 2022-08-29

**Authors:** Ruyang Han, Gengwei Yang, Deming Xu, Lu Jiang, Zhixiang Fu, Gang Zhao

**Affiliations:** 1The State Key Laboratory of Refractories and Metallurgy, Wuhan University of Science and Technology, Wuhan 430081, China; 2Institute for Frontier Materials Geelong, Deakin University, Waurn Ponds, VIC 3216, Australia

**Keywords:** high-strength ferritic steel, Ti−Mo−V complex microalloying, thermodynamics of precipitation, kinetics for precipitation, precipitation behavior

## Abstract

In this work, the precipitates in Ti−Mo−V steel were systematically characterized by high-resolution transmission electron microscopy (HRTEM). The thermodynamics and kinetics of precipitates in Ti−Mo and Ti−Mo−V steels were theoretically analyzed, and the effect of vanadium on the precipitation behavior was clarified. The results showed that the precipitation volume fraction of the Ti−Mo−V steel was significantly higher than that of Ti−Mo steel. The randomly dispersed precipitation and interphase precipitation (Ti, Mo, V)C particles coexisted in the Ti−Mo−V steel. When the temperature was higher than 872 °C, the addition of vanadium could increase the driving force for (Ti, Mo, V)C precipitation in austenite, resulting in an increased nucleation rate and shortened incubation period, promoting the (Ti, Mo, V)C precipitation. When the temperature was lower than 872 °C, the driving force for (Ti, Mo, V)C precipitation in austenite was lower than that for (Ti, Mo)C precipitation, and the incubation period of (Ti, Mo, V)C precipitation was increased. Moreover, it was also found that the precipitated-time-temperature curve of (Ti, Mo, V)C precipitated in the ferrite region was “C” shaped, but that of (Ti, Mo)C was “ε” shaped, and the incubation period of (Ti, Mo, V)C was significantly shorter than that of (Ti, Mo)C.

## 1. Introduction

With the development of industry, microalloyed high-strength steels have been widely used in transportation, automobile, construction, bridge and other engineering applications due to their excellent mechanical properties [1,2]. Microalloy and carbon atoms form nano-sized particles, which effectively hinder the movement of dislocations, thereby improving the strength of hot-rolled ferritic steels [3,4,5]. Among them, (Ti, Mo)C particles have a smaller size and higher coarsening resistance. Therefore, it is generally believed that Ti−Mo composite microalloying is more obvious to improve the properties of hot-rolled ferritic steels [6,7,8]. For instance, Funakawa et al. [9] indicated that the contribution of (Ti, Mo)C particles to yield strength was estimated to be over 300 MPa, and the yield strength of steel was significantly improved.

Recently, the literature [10,11,12,13,14,15,16] reported that adding Nb or V to Ti−Mo steel could further optimize microstructures and improve mechanical properties. Cai et al. [10] improved the yield strength of ultra-low carbon Ti−Mo steel to 680 MPa by adding Nb. Bu et al. [11] reported that the interphase precipitation in Ti−Mo−Nb steel could provide precipitation strengthening of ~320 MPa, resulting in its yield strength up to 747 MPa. Furthermore, our previous studies [12,13] also indicated that the addition of Nb on Ti−Mo hot-rolled ferritic steels could increase the contribution of precipitation strengthening. Meanwhile, the addition of Nb could also refine ferrite grains and inhibit bainitic transformation, thus improving the mechanical properties of hot-rolled ferritic steel. Compared to Nb, V could also inhibit bainitic transformation and has a stronger ability for precipitation strengthening [14]. Zhang et al. [15] reported that Ti−Mo−V composite microalloying could improve the yield strength of ferritic steel to 900 MPa due to the contribution of nano-sized (Ti, Mo, V)C particles. Fu et al. [16] found that the contribution of precipitation strengthening provided by a large number of nano-sized spherical (Ti, Mo, V)C particles was up to ~40% of the yield strength in ferritic steel. In conclusion, the precipitation strengthening provided by nano-sized (Ti, Mo, V)C particles is a key factor to further improve the strength of the Ti−Mo−V steel. However, the previous studies [10,11,12,13,14,15,16] of Ti−Mo−V steel mainly focused on phase the microstructure and mechanical properties. The precipitation mechanism of (Ti, Mo, V)C particles and the effect of V addition on precipitation behavior has not been studied systematically yet. An in-depth study is needed. This is the innovation of this paper.

Therefore, based on the previous work, this paper further provides the quantitative of precipitation in Ti−Mo and Ti−Mo−V steels by means of electrolytically extracted phase analysis, and the effect of V addition on the precipitation behavior was clarified through theoretical analyses. The results will provide a theoretical basis for the development and production of Ti−Mo−V high-strength steels. 

## 2. Materials and Experimental Procedure

The chemical compositions of the experimental steels are presented in Table 1. In total, 50 kg ingots were melted in a vacuum melting furnace and then forged into 250 mm × 100 mm × 60 mm billets. As shown in Figure 1, the billets were austenitized and homogenized at 1250 °C for 2 h and then hot-rolled to 3 mm in thickness via 7 passes with the finish rolling temperature of 860 °C. Subsequently, the hot-rolled sheets were cooled to 600 °C and held for 2 h to simulate the coiling process, followed by furnace cooling to ambient temperature.

The morphology and size of precipitates were characterized by a JEM-2100 transmission electron microscope (TEM) equipped with an energy dispersive spectroscope (EDS). The TEM samples were polished to less than 50 μm and punched into 3 mm discs. The discs were twin-jet polished with an electrolyte solution containing 6% perchloric acid and 94% ethyl alcohol. Subsequently, direction of the incident electron beam was adjusted parallel to the axis of the interphase precipitates band to observe the morphology of the interphase precipitates. Meanwhile, more than 800 precipitated particles were measured in the TEM images to obtain the average particle size and distribution. In addition, the mass fraction of precipitates was determined by electrolytically extracted phase analysis, and the test procedure was described in the reference [17]. 

The meanings and units of symbols in the thermodynamic and kinetic model are shown in Table 2.

## 3. Results and Discussion

### 3.1. Quantitative Analysis of Precipitation in Ti-Mo and Ti−Mo−V Steels 

In our previous works [13,14,15,16], it was found that the addition of V significantly improved the yield strength and tensile strength of Ti−Mo steel by ~28% and ~30%, respectively, mainly due to the increment of precipitation strengthening by (Ti, Mo, V)C precipitates. To further confirm the quantitative results of precipitates in Ti−Mo and Ti−Mo−V steels, electrolytically extracted phase analysis was carried out, and the results are shown in Table 3 and Table 4. It can be found that the Ti is almost completely precipitated in the two experimental steels, and the precipitation of Mo in Ti−Mo and Ti−Mo−V steel is similar. Nevertheless, the volume fraction of MC precipitates increases from 0.242% to 0.389% by addition of V. The precipitation of M_3_C is mainly controlled by the thermomechanical control process (TMCP) [14]. More C is consumed by forming MC particles in Ti−Mo−V steel, resulting in the mass fraction of M_3_C in Ti−Mo−V steel, which is less than that in Ti−Mo steel (0.391% vs. 0.416%).

### 3.2. Precipitation Characterization 

To clarify the morphology and position of precipitates in the matrix, the TEM observation of Ti−Mo−V steel is shown in Figure 2. The results show that a large number of precipitates exist in the steel. Meanwhile, the coarse particles (over 50 nm) mainly present two types, as shown in Figure 2a. The first type is large ellipsoidal particles (P1) that precipitated in the ferrite grains, which is identified as Ti-enriched (Ti, Mo, V) C particles (Figure 2c). The second type (P2) is verified as cementite mainly distributed in the grain boundaries. According to the previous studies [18,19], these (Ti, Mo, V)C particles are mainly precipitated during the soaking at high temperature.

In addition, nano-sized interphase precipitation is also observed in Figure 2b. It can be found that the precipitation spacing and morphology of the interphase precipitates change with the distance from the grain boundary. Yan et al. [20] pointed that the spacing of interphase precipitation was mainly related to the movements of the γ/α interface during the phase transformation. At the distance from the grain boundary, corresponding to the early stage of ferrite transformation, the phase transformation driving force is larger, resulting in faster movement of the γ/α interface. At this stage, the mechanism of interphase precipitation is a bowing mechanism, which leads to the irregular and larger particle spacing of interphase precipitation [20,21]. At the later stage of phase transformation, the γ/α interface moves slowly due to the decrease of phase transformation driving force, thereby changing to the quasi-step mechanism of interphase precipitation [22]. Then, the morphology of interphase precipitation changes to regular particle spacing in the (113) crystal planes of ferrite, and the interphase precipitation spacing decreases from 29.83 nm to 18.27 nm.

Another type of nano-sized particles is randomly dispersed precipitates. These carbides tend to precipitate at high energy sites, such as grain boundaries and dislocations. The bright-field image, dark-field image and electronic diffraction pattern of precipitates were observed by using the thin foil sample of experimental steel, as shown in Figure 3. The results indicate that the orientation relationship between (Ti, Mo, V)C precipitates and the ferrite matrix is clarified as ${\left(111\right)}_{\left(\mathrm{Ti},\;\mathrm{Mo},V\right)C}$//${\left(01\bar{1}\right)}_{\alpha -\mathrm{Fe}}$ and ${\left[01\bar{1}\right]}_{\left(\mathrm{Ti},\;\mathrm{Mo},V\right)C}$//${\left[\bar{1}11\right]}_{\alpha -\mathrm{Fe}}$ which agrees with the K−S relationship [14], and its lattice constant is calculated to be 0.423 nm. It means that these (Ti, Mo, V)C particles are mainly precipitated in the austenite during the rolling process. Meanwhile, the (Ti, Mo, V) C particles distributed in the trigeminal grain boundary can inhibit the movement of grain boundaries and refine the ferritic grains.

Figure 4 shows the Fourier transform of high-resolution image of precipitates with the size below 10 nm. It can be found that their orientation relationship agrees with the B−N relationship as ${\left(200\right)}_{\left(\mathrm{Ti},\;\mathrm{Mo},V\right)C}$//${\left(200\right)}_{\alpha -\mathrm{Fe}}$ and ${\left[011\right]}_{\left(\mathrm{Ti},\;\mathrm{Mo},V\right)C}$//${\left[001\right]}_{\alpha -\mathrm{Fe}}$ This indicates that these particles are precipitated in the ferrite, which can provide stronger precipitation strengthening.

### 3.3. Effect of V on the Precipitation Behavior in the Austenite of Ti−Mo Steel

According to the previous studies [23], the nano-sized carbides precipitated in the austenite refined the microstructure and strengthened the matrix. In addition, the precipitation behavior of carbides can be analyzed by thermodynamic and kinetic models [24]. It should be noted that the Ti easily forms TiN precipitates with lower solubility at a higher temperature. Additionally, the N in the experimental steel is an incidental element in the smelting. Thus, in order to facilitate the calculations, the N in experimental steel is assumed to be consumed completely, and the content of Ti in experimental steel is modified according to the stoichiometry of TiN in the relevant calculation. Furthermore, the content of the microalloying elements solid-solved in the austenite matrix at different temperatures can be calculated according to the solid solubility product formulae of TiC, MoC and VC.

Its formulae are as follows [14,15,24]
(1)$$\mathrm{lg}{\left\{[Ti]\cdot [C]\right\}}_{\gamma }=2.75-7000/T$$
(2)$$\mathrm{lg}{\left\{[Mo]\cdot [C]\right\}}_{\gamma }=4.251-3468/T$$
(3)$$\mathrm{lg}{\left\{[V]\cdot [C]\right\}}_{\gamma }=6.72-9500/T$$
(4)$$f_{v}=(\sum M_{i}-\sum \left[M_{i}\right]+C-[C])\frac{\rho_{Fe}}{100\rho_{MC}}$$
where the meanings and units of different symbols are shown in Table 2. The subscript γ indicates that the formula is applicable to the austenite matrix. The TiC, MoC and VC in the steel are all NaCl-type face-centered cubic structures, which can be mutually solved with each other. Therefore, the chemical formulae of (Ti, Mo)C and (Ti, Mo, V)C can be expressed as Ti*_x_*Mo*_y_*C (*x* + *y* = 1) and Ti*_x_*Mo*_y_*V*_z_*C (*x* + *y* + *z* = 1), respectively. Figure 5 shows the effects of temperature on the stoichiometric coefficient of Ti*_x_*Mo*_y_*C and Ti*_x_*Mo*_y_*V*_z_*C. The results show that the precipitates in Ti-Mo steel are mainly Ti-enriched (Ti, Mo)C, and the proportion of Ti is above 99.5%. For Ti−Mo−V steel, the precipitate is Ti-enriched (Ti, Mo, V)C at the higher temperature. However, with the decrease of temperature, the proportion of Ti in the (Ti, Mo, V)C decreases obviously, but the proportion of V increases. Meanwhile, the proportion of Mo is almost unchanged. This indicates that the Ti element has a high precipitation temperature, and the Mo element is basically not precipitated in austenite at high temperatures.

According to the thermodynamics calculation, the precipitation volume fractions of (Ti, Mo)C and (Ti, Mo, V)C as a function of temperature are obtained, as shown in Figure 6. The volume fraction of (Ti, Mo, V)C is significantly higher than that of (Ti, Mo)C and both of them increase with the decrease of temperature. When the temperature is lower than 900 °C, a large amount of V is precipitated and the volume fraction of (Ti, Mo, V)C increases rapidly, resulting in a larger difference between the volume fractions of (Ti, Mo, V)C and (Ti, Mo)C.

The kinetic model is based on the following assumptions: (1) MC particles nucleate on the dislocation line, and the nucleation rate rapidly decays to zero; (2) the edge dislocation is considered as the nucleation positions of precipitated particles; (3) the shape of precipitation is considered to spheroid. Then, the interfacial energy and driving force of (Ti, Mo)C and (Ti, Mo, V) can be expressed as [24,25,26]:(5)$$\sigma =\sum n\cdot \sigma_{M_{i}}$$
(6)$$\Delta G_{V}=\frac{1}{V_{m}}\left\{-19.1446B+19.1446T\left[A-\log\left(\prod {\left[M_{i}\right]}^{n}\left[C\right]\right)\right]\right\}$$
where the meanings and units of different symbols are shown in Table 2. The nucleation rate and precipitation start time of precipitates in different temperatures can be calculated by Formulas (7) and (8), respectively [15,24,25,26].
(7)$$I=K\cdot d^{*}{}^{2}\cdot \exp(\frac{\Delta G^{*}+Q}{kT})$$
(8)$$\mathrm{lg}(\frac{t_{0.05da}}{t_{0da}})=\left[-1.28994-2\mathrm{lg}d^{*}(\frac{1}{\ln10}\times \frac{(1+\frac{A\Delta G_{V}}{2\pi \sigma^{2}})\Delta G^{*}+\frac{5}{3}Q}{kT})\right]$$
where the meanings and units of different symbols are shown in Table 2.

The curves of interfacial energy, Gibbs free energy, nucleation rate-temperature (NrT) and precipitated-time-temperature (PTT) of the precipitates in Ti−Mo and Ti−Mo−V steels are shown in Figure 7. The results indicate that the interfacial energy and driving force of (Ti, Mo)C and (Ti, Mo, V)C increases with the decrease of temperature. When the temperature is higher than 902 °C, the interfacial energy of (Ti, Mo, V)C is similar to that of (Ti, Mo)C, and the driving force of (Ti, Mo, V)C is higher than that of (Ti, Mo)C, resulting in the NrT and PTT curves of (Ti, Mo, V)C shifting toward the top zone. Moreover, when the temperature is lower than 902 °C, the driving force for (Ti, Mo, V)C is significantly lower than that for (Ti, Mo)C, but the interfacial energy of (Ti, Mo, V)C/austenite is lower than that of (Ti, Mo)C/austenite. Thus, combined with the influence of driving force and interfacial energy on the precipitation, the precipitation of (Ti, Mo, V)C can be promoted above 872 °C.

### 3.4. Effect of V on the Precipitation Behavior in Ferrite of Ti−Mo Steel

It is well-known that the microalloying elements solubility in ferrite are significantly lower than those in austenite. The supersaturated solute microalloying elements will further precipitate in the ferrite during the coiling process. Compared to austenite, the precipitation precipitated in ferrite is smaller in size, thereby producing stronger precipitation strengthening [18,19,26]. It should be noted that the contents of solid solution of microalloying elements and carbon in the austenite at 860 °C are used as the initial content in the ferrite calculation model. The solution product formulas of TiC, MoC and VC in ferrite are as follows [18,19,27,28,29]:(9)$$\mathrm{lg}{\left\{[Ti]\cdot [C]\right\}}_{\alpha }=4.4-9575/T$$
(10)$$\mathrm{lg}{\left\{[Mo]\cdot [C]\right\}}_{\alpha }=6.163-7583/T$$
(11)$$\mathrm{lg}{\left\{[V]\cdot [C]\right\}}_{\alpha }=4.55-8300/T$$
where the meanings and units of different symbols are shown in the Table 2. The subscript α indicates that the formula is applicable to the ferrite matrix. Figure 8 shows the stoichiometric coefficient of Ti*_x_*Mo*_y_*C and Ti*_x_*Mo*_y_*V*_z_*C as a function of temperature in ferrite. The results show that as the temperature decreases, the proportion of Ti content decreases rapidly, whereas the proportion of Mo content increases. The precipitates in ferrite are Mo-enriched (Ti, Mo)C. For (Ti, Mo, V)C; with the decreases of temperature, the proportion of V decreases, but the proportion of Mo increases, whereas the proportion of Ti is basically unchanged. Meanwhile, the precipitation in the steel is mainly V-enriched (Ti, Mo, V)C particles, and the proportion of V content was above 85%. This indicates that V is the dominant element of (Ti, Mo, V)C in ferrite.

Figure 9 is the curve of the volume fraction of precipitation in ferrite with temperature. The results indicate that the volume fraction of precipitates increases with the decrease of temperature. The precipitation of V leads to the fact that the volume fraction of (Ti, Mo, V)C in ferrite is significantly higher than that of (Ti, Mo)C.

Different from austenite, the precipitates in ferrite mainly present rod-like and disc-like, and the orientation relationship between precipitation and ferritic matrix changes from a K−S relationship to a B−N relationship. The kinetic model in ferrite needs to introduce shape factors *η*, whereas *d^*^* and Δ*G^*^* are transformed into *d**_d_*^*^ and Δ*G**_d_*^*^. Therefore, Equations (7) and (8) have been changed to Equations (12) and (13), respectively [24,25,26].
(12)$$I=K\cdot d_{d}^{*}{}^{2}\cdot \exp(\frac{\Delta G_{e}^{*}+Q}{kT})$$
(13)$$\mathrm{lg}(\frac{t_{0.05da}}{t_{0da}})=\left[-1.28994-2\mathrm{lg}d_{d}^{*}(\frac{1}{\ln10}\times \frac{(1+\frac{\eta A\Delta G_{V}}{2\pi \sigma^{2}})\Delta G_{e}^{*}+\frac{5}{3}Q}{kT})\right]$$
where the meanings and units of different symbols are shown in Table 2.

Figure 10 presents the interfacial energy, Gibbs free energy, NrT and PTT curves of (Ti, Mo)C and (Ti, Mo, V)C precipitated in the ferrite. It is found that PTT curve of (Ti, Mo, V)C precipitated in the ferrite region is “C” shaped, but that of (Ti, Mo)C is “ε” shaped. The driving force for (Ti, Mo, V)C is higher than that for (Ti, Mo)C, and the interfacial energy of (Ti, Mo, V)C/ferrite is lower than that of (Ti, Mo)C/ferrite, resulting in the nucleation rate of (Ti, Mo, V)C in ferrite being significantly larger than that of (Ti, Mo)C. Consequently, the precipitation of (Ti, Mo, V)C in ferrite is greatly promoted, and improved the precipitation strengthening of Ti−Mo−V hot-rolled ferritic steel.

## 4. Conclusions

In this study, the precipitates of Ti−Mo−V microalloyed high-strength steel were characterized in detail. The effect of V addition on precipitation behavior in austenite and ferrite was explored. The main conclusion could be summarized as follows:
The addition of V can significantly increase the volume fraction of the (Ti, Mo, V)C precipitates in Ti−Mo−V steel (0.242% vs. 0.389%). The precipitation characterization shows that the (Ti, Mo, V)C particles can be divided into three types: spherical precipitates in austenite after deformation, interphase precipitates during γ→α transformation and dispersive nano-sized precipitates in the supersaturated ferrite matrix.The results of theoretical calculation indicate that when the temperature is higher than 872 °C, the addition of vanadium can increase the driving force for (Ti, Mo, V)C precipitation in austenite, resulting in an increased nucleation rate and shortened incubation period, promoting the (Ti, Mo, V)C precipitation.The PTT curve of (Ti, Mo, V)C precipitated in the ferrite region is “C” shaped, but that of (Ti, Mo)C is “ε” shaped, and the incubation period of (Ti, Mo, V)C is significantly shorter than that of (Ti, Mo)C.

## Figures and Tables

**Figure 1 materials-15-05965-f001:**
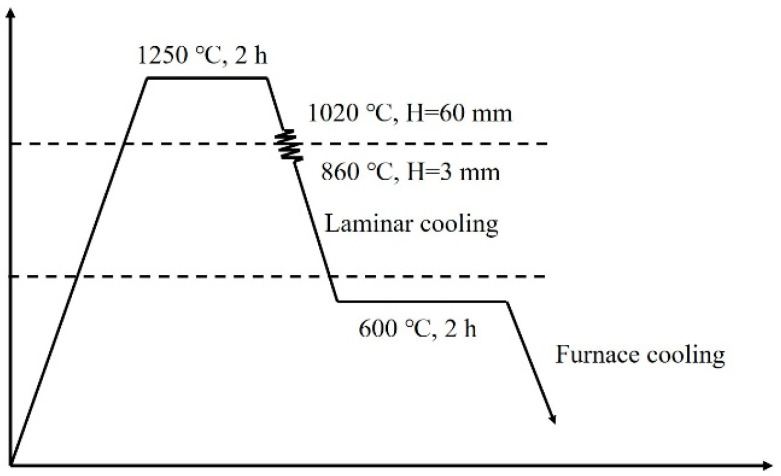
Schematic diagram of rolling process.

**Figure 2 materials-15-05965-f002:**
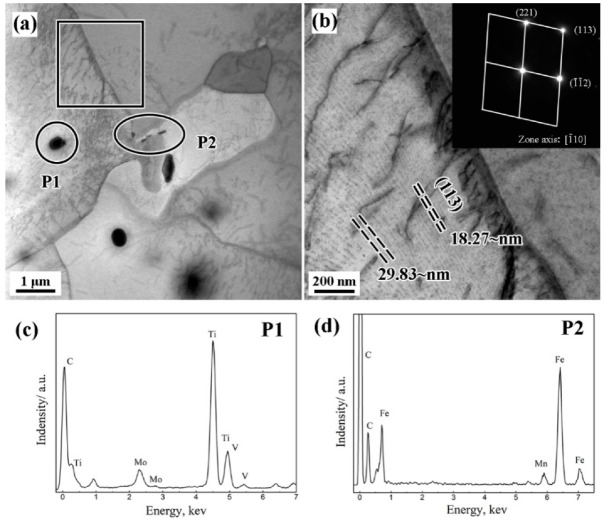
TEM and EDS images of particles in Ti−Mo−V steel. (**a**) TEM morphologies; (**b**) Interphase precipitation morphologies; (**c**) EDS analysis result of P1; (**d**) EDS analysis result of P2.

**Figure 3 materials-15-05965-f003:**
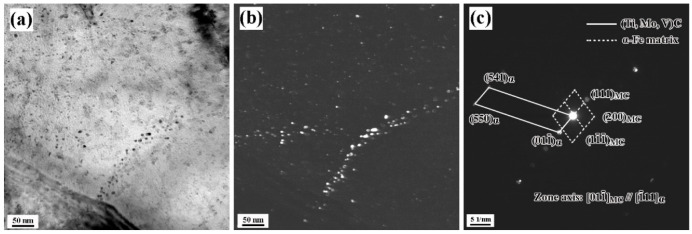
The (**a**) bright-field image (**b**), dark-field image of precipitates and (**c**) diffraction pattern of precipitates.

**Figure 4 materials-15-05965-f004:**
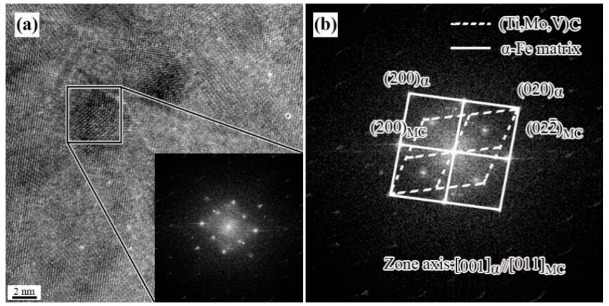
(**a**) High resolution TEM micrograph of the precipitation and (**b**) fast Fourier transformed.

**Figure 5 materials-15-05965-f005:**
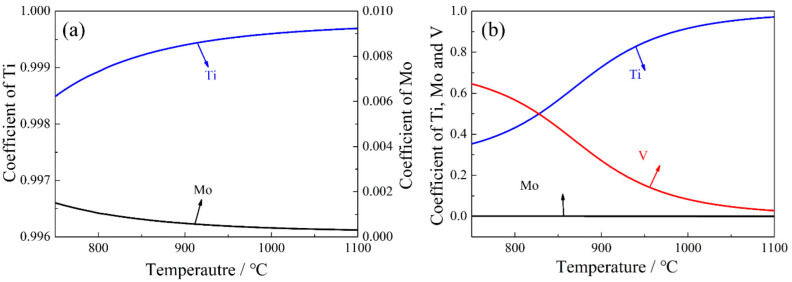
Effects of temperature on the formula coefficient of (**a**) Ti*_x_*Mo*_y_*C and (**b**) Ti*_x_*Mo*_y_*V*_z_*C in austenite.

**Figure 6 materials-15-05965-f006:**
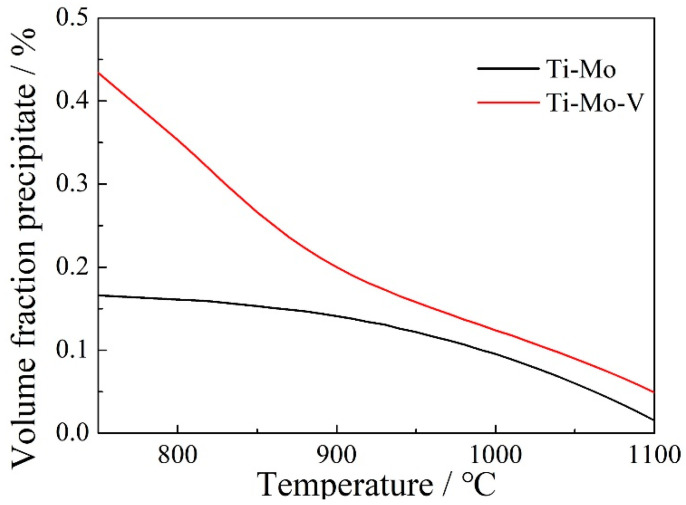
Volume fractions of precipitation in Ti−Mo and Ti−Mo−V steels as a function of temperature in austenite.

**Figure 7 materials-15-05965-f007:**
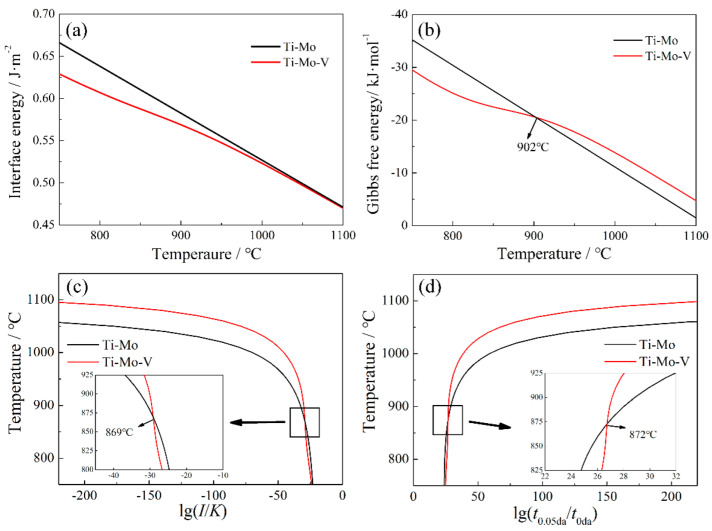
The kinetic analysis in austenite of Ti−Mo and Ti−Mo−V steels. (**a**) curve of interfacial energy, (**b**) curve of Gibbs free energy, (**c**) curve of NrT and (**d**) curve of PTT.

**Figure 8 materials-15-05965-f008:**
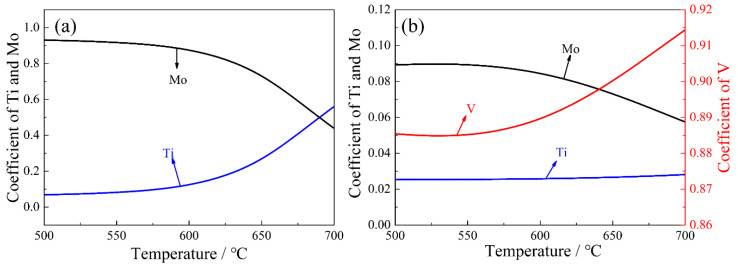
The formula coefficient of (**a**) Ti*_x_*Mo*_y_*C and (**b**) Ti*_x_*Mo*_y_*V*_z_*C in ferrite as a function of temperature.

**Figure 9 materials-15-05965-f009:**
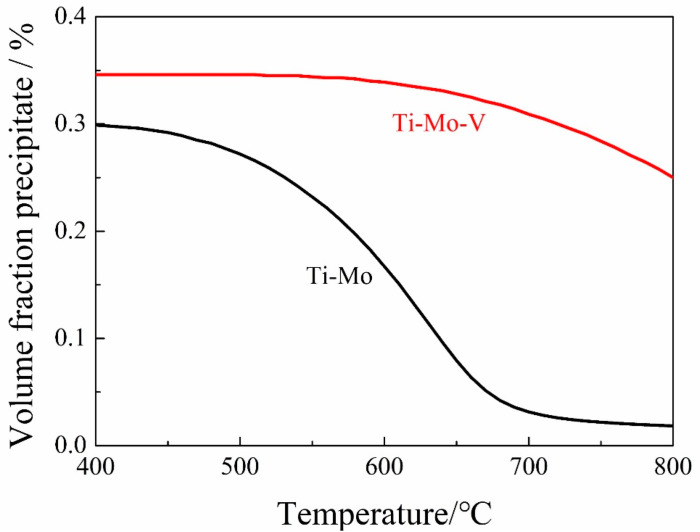
Volume fractions of precipitation in Ti−Mo and Ti−Mo−V steels as a function of temperature in ferrite.

**Figure 10 materials-15-05965-f010:**
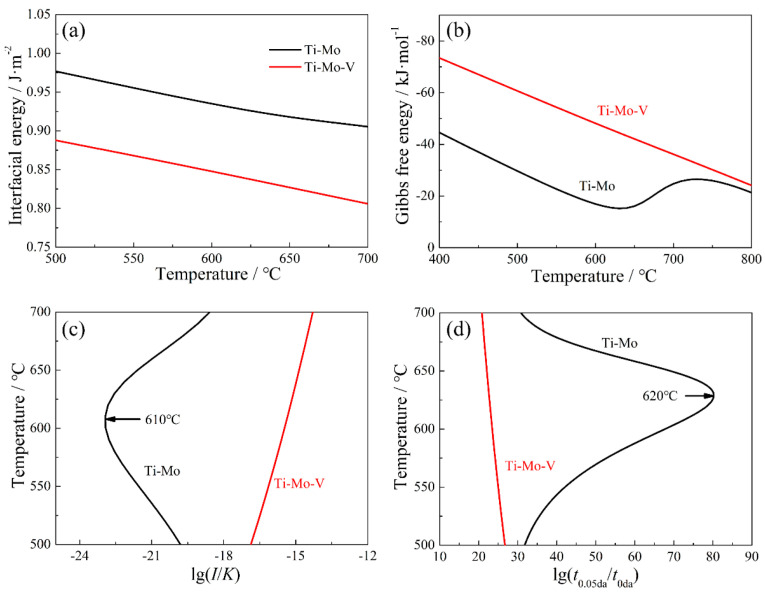
The thermodynamic and kinetic analysis in ferrite of Ti−Mo and Ti−Mo−V steels. (**a**) curve of interfacial energy, (**b**) curve of Gibbs free energy, (**c**) curve of NrT and (**d**) curve of PTT.

**Table 1 materials-15-05965-t001:** Chemical compositions of experimental steels (wt.%).

Steel	C	Si	Mn	Ti	Mo	V	Cr	N	Fe
Ti−Mo	0.06	0.07	1.44	0.097	0.28	-	0.21	0.0035	Bal.
Ti−Mo−V	0.08	0.14	1.48	0.10	0.30	0.24	0.22	0.0049	Bal.

**Table 2 materials-15-05965-t002:** Meanings and units of different symbols.

Symbol	Meaning	Unit
*M_i_*	Amount of microalloyed element in the steel	wt.%
[*M_i_*]	Amount of solid solution of element *M_i_*	wt%
*f* _v_	Volume fraction of precipitation	%
*ρ* _Fe,_ *ρ* _MC_	Density of Fe and MC precipitated particles	kg/m^3^
*K*	Temperature-independent constant	-
*t* _0da_	Temperature-independent parameter	s
*t* _0.05da_	Start time of precipitation that corresponds to the fraction precipitates 5%	s
*d* ^*^	Size of critical nucleus in austenite	nm
*d_d_* ^*^	Size of critical nucleus in ferrite	nm
*A*	Core energy of an edge dislocation line per unit	J/m
*σ*	Interfacial energy	J/m^2^
$\sigma_{M_{i}}$	Specific interfacial energy between *M_i_C* and matrix	J/m^2^
Δ*G*_V_	Volume free energy	J/m^3^
Δ*G*^*^	Critical nuclear power in austenite	J
Δ*G_d_*^*^	Critical nuclear power in ferrite	J
*V* _m_	Molar volume of precipitates	m^3^/mol
*A*, *B*	Constants of precipitates in the solubility product formula	-
*n*, *x*, *y*, *z*	Stoichiometric coefficient	-
*η*	Shape factor	-
*Q*	Activation energy for atoms	J/mol
*k*	Boltzmann constant	-

**Table 3 materials-15-05965-t003:** Quantitative analysis of MC precipitates of Ti−Mo and Ti−Mo−V steel.

Steel	Mass Fraction of Element in MC Phase/%					*f*_v_/%
	Ti ^a^	Mo	V	C ^b^	∑	
Ti−Mo	0.082	0.085	-	0.031	0.212	0.242
Ti−Mo−V	0.077	0.087	0.086	0.050	0.300	0.389

^a^ Mass fraction of Ti in TiN was deducted. ^b^ Carbon content was an estimated result according to MC formula.

**Table 4 materials-15-05965-t004:** Quantitative analysis of M_3_C precipitates of Ti−Mo and Ti−Mo−V steel.

Steel	Mass Fraction of Element in M_3_C Phase/%					
	Fe	Mn	Mo	V	C ^c^	∑
Ti−Mo	0.323	0.033	0.026	-	0.027	0.416
Ti−Mo−V	0.299	0.018	0.031	0.01	0.025	0.391

^c^ Carbon content was an estimated result according to M_3_C formula.

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
