# Peer review of "Effect of V on the Precipitation Behavior of Ti−Mo Microalloyed High-Strength Steel"

_materials, 2022, doi:10.3390/ma15175965_

Round 1

Reviewer 1 Report

shown on manuscript

Author Response

Dear reviewer,

We appreciate your professional comments on our manuscript. This gives us an opportunity to improve the quality of our work. We have made all necessary revisions according to your comments and revised references accordingly. We have carefully checked the full paper, and modified places are highlighted in blue in the revised manuscript. Detailed explanations of the revisions are given as follows point by point.

Comment: Since this analysis is quantitative you need to provide a short analysis of the uncertainty of the analysis itself and the measurements which lead to the analysis.

Response: The electrolytically extracted phase analysis is widely used to investigate the second phase particles in the microalloyed steel. It can provide a quantitatively analysis on the compositional composition and crystal structure of the second phase particles. And the experimental procedure is described in the reference [17].

Previous studies have pointed out that the MC particles have a lower solid solubility than M3C particles, resulting in preferential precipitation of MC during the cooling process. The addition of V on Ti-Mo steel leads to more C being consumed to form (Ti, Mo, V)C particles. Thus, the C content of M3C particles is insufficient in Ti-Mo-V steel, resulting in a reduction of mass fraction of M3C.

The sentence “The two steels have the same parameters of TMCP, resulting in the similar mass fractions of M3C in Ti-Mo and Ti-Mo-V steels (0.416% vs 0.391%).” has been changed to “More C is consumed by forming MC particles in Ti-Mo-V steel, resulting in the mass fraction of M3C in Ti-Mo-V steel is slightly less than that in Ti-Mo steel (0.391% vs0.416%).”.

[17] G. Yang, X. Sun, Z. Li, X. Li, Q. Yong, Effects of vanadium on the microstructure and mechanical properties of a high strength low alloy martensite steel, Mater. Des. 2013, 50, 102-107.

Reviewer 2 Report

·       There are many articles of titanium alloys. The following references are recommended on Titanium group: [1] Characterization and Mechanical Proprieties of New TiMo Alloys Used for Medical Applications; [2] New Titanium Alloys, Promising Materials for Medical Devices; [3] Cytocompatibility of pure metals and experimental binary titanium alloys for implant materials; [4] Ti based biomaterials, the ultimate choice for orthopaedic implants - A review

·       Show the novelty of the paper compared to the literature, however there is still much work on this topic.

·       Why you choose these alloys? Discuss also by applicability. This alloys can be used like biomaterials.

·       In the Introduction section, the last paragraph should contain the scientific contribution and scientific hypotheses of your research. Complete, further elaborate the scientific contribution and scientific hypotheses of your research. Be explicit. In addition to the goal of the research (which was written), the novelty in the context of the scientific contribution should be pointed out. Scientific contributions should be written based on the shortcomings of previous research in the literature. In this way, the authors will better emphasize novelty and scientific soundness.

·       Analyze and discuss possibilities of practical application.

·       Complete the conclusions with the limitations of the proposed methodology. Also write future research.

·       Generally, the quality of the writing could be improved.

Author Response

Dear reviewer,

We appreciate your professional comments on our manuscript. This gives us an opportunity to improve the quality of our work. We have made all necessary revisions according to your comments and revised references accordingly. The modified places are highlighted in blue in the revised manuscript. Detailed explanations of the revisions are given as follows point by point.

Comment (1): There are many articles of titanium alloys. The following references are recommended on Titanium group: [1] Characterization and Mechanical Proprieties of New TiMo Alloys Used for Medical Applications; [2] New Titanium Alloys, Promising Materials for Medical Devices; [3] Cytocompatibility of pure metals and experimental binary titanium alloys for implant materials; [4] Ti based biomaterials, the ultimate choice for orthopaedic implants - A review.

Response: Thanks to your advice. However, the material in this study is an Fe-based Ti-Mo-V microalloy material. The experimental materials in the above references are the Ti-based alloy materials, which are not consistent with the materials in this study. According to your comment, the sentence “Microalloy and carbon atoms could form nano-sized particles, which effectively hindered the movement of dislocations, thereby improving the strength of hot-rolled ferritic steels [3-5].” has been added to the paper. And the new references [3-5] have been added in the revised paper

[3] Liu, C. F. Xiong, Y. Wang, Y. Cao, X. Liu, Z. Xue, Q. Peng, L. Peng, Strengthening Mechanism and Carbide Precipitation Behavior of Nb-Mo Microalloy Medium Mn Steel. Materials 2021, 14, 7461.

[4] J. Yuan, Y. Xiao, N. Min, W. Li, S. Zhao, The Influence of Precipitate Morphology on the Growth of Austenite Grain in Nb-Ti-Al Microalloyed Steels. Materials 2022, 15, 3176.

[5] X. Li, J. Yang, Y. Li, L. Liu, C. Jin, X. Gao, X. Deng, Z. Wang, A Systematical Evaluation of the Crystallographic Orientation Relationship between MC Precipitates and Ferrite Matrix in HSLA Steels. Materials 2022, 15, 3967.

Comment (2): Show the novelty of the paper compared to the literature, however there is still much work on this topic.

Response: The previous works reported that adding V to Ti-Mo steel could further improve the mechanical properties, and precipitation strengthening provided by nano-sized (Ti, Mo, V)C particles is a key factor to further improve the strength. However, the previous studies of Ti-Mo-V steel mainly focused on phase the microstructure and mechanical properties. The precipitation mechanism of (Ti, Mo, V)C particles and the effect of V addition on the precipitation behavior has not been studied systematically yet. An in-depth study needed. This is the innovation of this paper. The sentences “However, the previous studies of Ti-Mo-V steel mainly focused on phase the microstructure and mechanical properties. The effect of V addition on the precipitation behavior has not been studied systematically yet. An in-depth study needed.” has been changed to “In conclusion, the precipitation strengthening provided by nano-sized (Ti, Mo, V)C particles is a key factor to further improve the strength of the Ti-Mo-V steel. However, the previous studies [10-16] of Ti-Mo-V steel mainly focused on phase the microstructure and mechanical properties. The precipitation mechanism of (Ti, Mo, V)C particles and the effect of V addition on the precipitation behavior has not been studied systematically yet. An in-depth study needed. This is the innovation of this paper.

Comment (3): Why you choose these alloys? Discuss also by applicability. This alloys can be used like biomaterials.

Response: The experimental material in this paper is microalloyed ferritic steel, which is mainly used in transportation, automobile, construction, bridge and other engineering applications. It is not currently applied to biomaterials.

Microalloying elements such as Ti, Mo, Nb and V combine with C to form nanoscale carbides that effectively impede dislocation movement, resulting in precipitation strengthening. According to the previous literatures, (Ti, Mo)C particles have a smaller size and higher coarsening resistance. Therefore, it is generally believed that Ti-Mo composite microalloying is more obvious to improve the properties of hot-rolled ferritic steels. In our previous studies, we found that the adding V to Ti-Mo steel could further increase the precipitation strengthening. Therefore, Ti-Mo-V microalloyed steel is chosen as the experimental material in this paper.

Comment (4): In the Introduction section, the last paragraph should contain the scientific contribution and scientific hypotheses of your research. Complete, further elaborate the scientific contribution and scientific hypotheses of your research. Be explicit. In addition to the goal of the research (which was written), the novelty in the context of the scientific contribution should be pointed out. Scientific contributions should be written based on the shortcomings of previous research in the literature. In this way, the authors will better emphasize novelty and scientific soundness. Analyze and discuss possibilities of practical application. Complete the conclusions with the limitations of the proposed methodology. Also write future research.

Response: Thanks for the reviewer's advice. The last paragraph in the introduction “Therefore, in this paper, the quantitative and theoretical analyses of precipitation in Ti-Mo and Ti-Mo-V steels were conducted, and the effect of V addition on the precipitation behavior was charified.” has been changed to “Therefore, based on the previous work, this paper further provides the quantitative of precipitation in Ti-Mo and Ti-Mo-V steels by means of electrolytically extracted phase analysis, and the effect of V addition on the precipitation behavior was clarified through theoretical analyses. The results will provide a theoretical basis for the development and production of Ti-Mo-V high-strength steels.

Reviewer 3 Report

Research topics are relevant. Research methods are selected correctly and described in detail. However, there are a number of comments that require corrections and additions.

1.     I disagree with the title of the article. Low-alloy steels according to state diagrams, depending on temperature, can have a microstructure of alloyed austenite or alloyed ferrite. When we talk about ferritic steels, we mean high alloy steels of the ferric class, for which the ferritic structure is maintained from room temperature to the melting temperature. Therefore, the title of the article can be the following «Effect of V on the precipitation behavior of Ti-Mo microalloyed 2 high-strength steel»

2.     It is not clear why during "rolling process" heating is carried out to a very high temperature – to 12500С. Based on the schematic diagram shown in Figure 1, the plastic deformation process occurs at a lower temperatures. Such overheating leads to the growth of austenite grains and, as a consequence, causes brittleness of the steel.

3.     Taking into account the chemical composition of the studied steels and greater chemical affinity of nitrogen to titanium, molybdenum, and vanadium than the affinity of carbon for these elements, we can assume that it is a possibility formation not pure carbides, but carbonitrides.

4.     I think that conclusion 1 and 2 should be combined. Since in the first conclusion, the authors describe different types of carbides that are identified in austenite, according to grain boundaries and in a ferritic matrix. Here we are talking about the same carbides that originate in austenite and precipitate along grain boundaries or on dislocations, or along the grain body of alloyed ferrite.

Author Response

Dear reviewer,

We appreciate your professional comments on our manuscript. This gives us an opportunity to improve the quality of our work. We have made all necessary revisions according to your comments and revised references accordingly. The modified places are highlighted in blue in the revised manuscript. Detailed explanations of the revisions are given as follows point by point.

Comment (1): I disagree with the title of the article. Low-alloy steels according to state diagrams, depending on temperature, can have a microstructure of alloyed austenite or alloyed ferrite. When we talk about ferritic steels, we mean high alloy steels of the ferric class, for which the ferritic structure is maintained from room temperature to the melting temperature. Therefore, the title of the article can be the following “Effect of V on the precipitation behavior of Ti-Mo microalloyed high-strength steel”.

Response: Thanks for your advice. The title of the article “Effect of V on the precipitation behavior of Ti-Mo microalloyed high-strength ferritic steel” has been changed to “Effect of V on the precipitation behavior of Ti-Mo microalloyed high-strength steel

Comment (2): It is not clear why during "rolling process" heating is carried out to a very high temperature – to 1250 °Ð¡. Based on the schematic diagram shown in Figure 1, the plastic deformation process occurs at a lower temperatures. Such overheating leads to the growth of austenite grains and, as a consequence, causes brittleness of the steel.

Response: The high-temperature heat process (1250 °C for 2 h) of billet is used in this study to simulate the industrial production process. In industrial production, high-temperature heat treatment improves the plasticity and reduces deformation resistance of billets, and it will reduce equipment accidents caused by wear and impact. Therefore, a larger rolling reduction can be adopted to improve productivity efficiency of the rolling mill. In addition, the high-temperature reheating process of the billet can completely homogenize alloy elements, thereby avoiding serious segregation.

Comment (3): Taking into account the chemical composition of the studied steels and greater chemical affinity of nitrogen to titanium, molybdenum, and vanadium than the affinity of carbon for these elements, we can assume that it is a possibility formation not pure carbides, but carbonitrides.

Response: Indeed, the chemical affinity of N for Ti, Mo and V is greater than that of C for these elements. For experimental steel, the N is incidental element with less content. And TiN has small solid solution in steel, and it is easily precipitated during high-temperature stage. Thereby, most of the N has been combined with Ti to form TiN particles. In the subsequent precipitation process, retained N forms to C-riched carbonitrides with Ti, Mo and V. However, the N content in the carbonitrides is very small, and there is no significant N peak in ESD analysis. Therefore, in order to facilitate the calculations, we assume the precipitated particles as pure carbides.

The sentences “It should be noted that the Ti easily forms TiN precipitates with lower solubility at a higher temperature. And the N in the experimental steel is an incidental element in the smelting. Thus, in order to facilitate the calculations, the N in experimental steel is assumed to be consumed completely, and the content of Ti in experimental steel is modified according to the stoichiometry of TiN in the relevant calculation. And the content of the microalloying elements solid-solved in the austenite matrix at different temperatures can be calculated according to the solid solubility product formulae of TiC, MoC and VC.” have been updated in the revised manuscript.

Comment (4): I think that conclusion 1 and 2 should be combined. Since in the first conclusion, the authors describe different types of carbides that are identified in austenite, according to grain boundaries and in a ferritic matrix. Here we are talking about the same carbides that originate in austenite and precipitate along grain boundaries or on dislocations, or along the grain body of alloyed ferrite.

Response: In our opinion, the three conclusions correspond to each of the three parts in the study. The conclusion 1 corresponds to the quantitative analysis results and precipitation characterization. And the second conclusion corresponds to the theoretical calculation of precipitation in austenite. The conclusion section has been changed as follow:

“(1) The addition of V can significantly increase the volume fraction of the (Ti, Mo, V)C precipitates in Ti-Mo-V steel (0.242% vs 0.389%). The precipitation characterization shows that the (Ti, Mo, V)C particles can be divided into three types: spherical precipitates in austenite after deformation, interphase precipitates during γ→α transformation and dispersive nano-sized precipitates in supersaturated ferrite matrix

(2) The results of theoretical calculation indicate that when the temperature is higher than 872 °C, the addition of vanadium can increase the driving force for (Ti, Mo, V)C precipitation in austenite, resulting in increased nucleation rate and shortened incubation period, promoting the (Ti, Mo, V)C precipitation.”

Round 2

Reviewer 2 Report

Paper was improved.

Reviewer 3 Report

I thank the authors for the reasoned responses to the comments and the additions and changes made to the article. I think that in this form the article can be published.